# Products of *Lactobacillus*
*delbrueckii* subsp. *bulgaricus* Strain F17 and *Leuconostoc*
*lactis* Strain H52 Are Biopreservatives for Improving Postharvest Quality of ‘Red Globe’ Grapes

**DOI:** 10.3390/microorganisms8050656

**Published:** 2020-04-30

**Authors:** Xiang Fang, Qinchun Duan, Zhuo Wang, Fuyun Li, Jianxiong Du, Wencan Ke, Diru Liu, Ross C. Beier, Xusheng Guo, Ying Zhang

**Affiliations:** 1College of Public Health, Lanzhou University, Lanzhou 730000, China; fangx17@lzu.edu.cn (X.F.); zhwang2018@lzu.edu.cn (Z.W.); lify17@lzu.edu.cn (F.L.); liudiru@lzu.edu.cn (D.L.); 2State Key Laboratory of Grassland and Agro-Ecosystems, School of Life Sciences, Lanzhou University, Lanzhou 730000, China; duanqch16@lzu.edu.cn (Q.D.); dujx16@lzu.edu.cn (J.D.); kewc12@lzu.edu.cn (W.K.); guoxsh07@lzu.edu.cn (X.G.); 3United States Department of Agriculture, Agricultural Research Service, Southern Plains Agricultural Research Center, Food and Feed Safety Research Unit, College Station, TX 77845-4988, USA; ross.beier@yahoo.com

**Keywords:** biopreservative, grapes, *Lactobacillus delbrueckii* subsp. *bulgaricus* strain F17, *Leuconostoc lactis* strain H52, postharvest, products

## Abstract

‘Red Globe’ table grapes are large, edible, seeded fruit with firm flesh that tastes good, but can have poor postharvest shelf-life. This study was conducted to explore the effects of products of *Lactobacillus delbrueckii* subsp. *bulgaricus* strain F17 and *Leuconostoc lactis* strain H52 on ‘Red Globe’ table grapes for the enhancement of shelf-life and improvement of grape quality characteristics during postharvest storage. Strains F17 and H52 were isolated from traditional fermented yak milk obtained in the Qinghai–Tibetan Plateau. Samples from untreated and treated grapes were analyzed for physicochemical, biochemical, and microbiological properties (weight loss, decay rate, pH, total soluble solids content, titratable acidity, total phenols, sensory evaluation, and microbial growth) for 20 days. The results demonstrated that supernatants from both strains significantly reduced weight loss, decay rate, aerobic mesophilic bacteria, and coliform bacteria counts; delayed maturity and senescence of table grapes; and reduced titratable acidity and total phenols. However, the supernatant of strain F17 was more effective and resulted in better sensory evaluations and had a significant inhibitory effect on yeast and molds by day 5. Meanwhile, the supernatant from strain H52 had a significant inhibitory effect on fungi over the whole storage period. In addition, the results of the Pearson correlation analysis suggested that weight loss, decay rate, total soluble solids content, and microorganisms were highly correlated with the sensory evaluation data and quality of postharvest grapes when treated with the products of strain F17. On the basis of these data and sensory organoleptic qualities, the supernatant containing products from strain F17 had the best potential as a biopreservative to improve the postharvest quality of ‘Red Globe’ table grapes.

## 1. Introduction

‘Red Globe’ table grapes are large, edible, seeded fruit with firm flesh that tastes good, but can have poor postharvest shelf-life. Fruits and vegetables are recommended as a regular part of the human diet because they contain antioxidants, dietary fiber, minerals, and vitamins, as well as other beneficial substances that help support good human health [1]. Nevertheless, most fruits are susceptible to physical injury and microbial invasion, resulting in a short postharvest life. Therefore, our laboratory has been studying the problem of how to extend the shelf-life and improve the quality of postharvest fruit and vegetables. With this question in mind, we selected ‘Red Globe’ table grapes to investigate ways of improving their postharvest shelf-life.

Table grapes are not only delectable and nutritious, but also a source of polyphenols and resveratrol, with high economic and dietary value [2]. Polyphenols are important bioactive substances and are products of grape berries, which play an important role in grape flavor; color; taste; and many probiotic functions such as anti-oxidation, anti-cancer, and prevention of cardiovascular and cerebrovascular diseases [3]. Moreover, resveratrol from grapes also has anti-inflammation and anticancer properties [4]. These characteristics have caused added interest for daily consumption, resulting in market demand of high-quality table grapes [5]. However, table grapes are a highly perishable, non-climacteric fruit that undergoes a variety of postharvest deteriorations including weight loss, softening, loss of nutritional, and functional compounds either at room or refrigerated temperatures during storage [6]. Furthermore, owing to a high sugar content, water content, thin skin, soft flesh, and susceptibility to infections by pathogenic bacteria, grapes are extremely perishable during storage, transportation, and sales, which greatly reduces the quality and commodity value of grapes.

The quality of postharvest table grapes in China and abroad is widely attempted to be maintained using sulfur dioxide (SO_2_) fumigation [7]. Moreover, numerous studies have shown that SO_2_ fumigation can significantly inhibit most common pathogens that cause grape decay, such as gray mold and *Rhizopus nigricans* [8,9]. However, the residue concentration of SO_2_ is one key factor used to determine the quality of postharvest grapes, and the levels of residues are difficult to control during the fumigation treatment process. Once SO_2_ fumigation leads to improper levels of SO_2_ residues, these residues can both be potentially harmful to human health and give a sulfurous flavor to the table grapes [10]. As consumer awareness of food contamination and human health has increased, most consumers are more aware of food safety issues and are critical of the use of chemical preservatives, as well as the preferable preservation technologies that will keep foods safe and free of chemicals [10,11]. In addition, coating table grapes with ‘aloe vera gel’ could reduce the loss of phenolics and ascorbic acid, resulting in a higher retention of total antioxidant activity, which improved the overall preservation of grape berries during cold storage and extended the shelf-life of grapes [12]. Therefore, some researchers have turned their attention to safe, pollution-free, environmentally friendly preservation technologies. It is worth noting that biological preservation technologies, as an alternative to chemical fungicides, have been recognized as promising methods to control postharvest diseases in fruits and vegetables [13]. The biological preservation technologies are often applied as edible antimicrobial coatings or the addition of natural antibacterial substances on the surface of fruits [14,15,16].

As it is possible to inhibit microbial growth by introducing other competitive microorganisms that are beneficial to humans, some microorganisms with biological protective effects have been proposed as biological preservatives [10]. *Lactobacillus* is a bacterium found in the natural microbiome of meat, milk, vegetables, and fish, and is used both as a protective culture and a producer of bacteriocins [17], and it has a long history of safe use in food. Moreover, *Lactobacillus* are generally recognized as safe (GRAS) by the U.S. Food and Drug Administration. Therefore, *Lactobacillus delbrueckii* subsp. *bulgaricus* strain F17 and *Leuconostoc lactis* strain H52, which were isolated from traditional fermented yak milk obtained in the Qinghai–Tibetan Plateau, were selected for this experiment. On the basis of previous studies in our laboratory, strains F17 and H52 possess favorable antioxidant activities. In addition, it was discovered that F17 produces a class IIa bacteriocin and H52 produces exopolysaccharides [18,19]. Subsp. *bulgaricus* is a lactic acid producing bacterium [20,21], and is acid tolerant [22]. It was demonstrated that F17 and H52 can significantly inhibit foodborne pathogens [18,19] such as *Salmonella typhi*, *Staphylococcus aureus*, *Listeria monocytogenes*, and *Escherichia coli* [23].

Although *Lactobacillus* and their metabolites are widely used to control foodborne pathogens and spoilage organisms in meat products to ensure food quality and safety [24,25,26,27], to the best of our knowledge, the application of products of lactic acid bacteria to improve table grape quality during storage at 25 °C has not been reported. Therefore, the focus of this study was on extending the shelf-life of fresh table grapes using a supernatant of *Lactobacillus* containing products as a potential biopreservative. *Lactobacillus* supernatant-treated grapes were compared to the untreated control grapes in terms of weight loss, decay rate, stem browning, pH, total soluble solids content (SSC), titratable acidity (TA), total phenols (TP), microbial counts, and sensory evaluation during storage at 0, 5, 10, 15, and 20 d at 25 °C. The overall effects of the supernatants of strains F17 and H52 were evaluated on the preservation of ‘Red Globe’ table grapes.

## 2. Materials and Methods

### 2.1. Raw Material

Ripe ‘Red Globe’ table grapes were harvested from a local orchard in Yuzhong County, Gansu province, China. The grapes were immediately transported to the laboratory following collection. Grapes with defects such as decay and mechanical damage were discarded and grape berries of uniform size, shape, and color were selected for the experiment. A total of 600 grape berries were randomly divided into the control, and F17 and H52 supernatant-treatment groups. The F17 and H52 grape berry groups were immersed in a fermentation supernatant (see Section 2.2) containing products from strain F17 or strain H52 for 2s. Meanwhile, untreated grapes served as the control. Clean plastic boxes containing holes were used to store 10 air-dried grape berries per box at 25 °C.

### 2.2. Preparation of Strains F17 and H52 Fermentation Supernatants

The frozen strains F17 and H52 were removed from a –80 °C ultra-low temperature freezer (DW-HL668, Zhongke MeiLing Cryogenics Company, Ltd., Hefei, China) and thawed at room temperature, and then cultured in Man–Rogosa–Sharpe (MRS) broth (Qingdao Hope Bio-Technology Co., Ltd., Qingdao, China) and incubated at 37 °C for 24 h. The bacteria were transferred to fresh MRS broth and incubated at 37 °C for 24 h, then the bacteria count of both F17 and H52 broths was adjusted to 1 × 10^8^ colony forming units (CFU)/mL using a UV/vis double beam U-2910 spectrophotometer (Hitachi, Tokyo, Japan). After that, the fermentation supernatant was separated from the bacteria by centrifugation at 8000× *g* for 15 min at 4 °C with a cryogenic high-speed Allegra 64R centrifuge (Beckman Coulter, Brea, CA, USA). The bacterial precipitate was discarded and the supernatants containing the products of strains F17 and H52 were stored at 4 °C until use.

### 2.3. Determination of Weight Loss

Weight loss of the table grapes in each group during storage was measured by monitoring the weight change of the grape berries after 0, 5, 10, 15, and 20 d of storage at 25 °C. Weight loss was calculated as the percentage loss of initial weight [28].

### 2.4. Decay Assessment

During storage, decay incidence was monitored by determining the natural microorganisms associated with the grape berries, and the disease severity of each grape berry was evaluated according to the following empirical scale [29]: ‘0’ = intact grape berry; ‘1’ = one lesion less than 2 mm in diameter; ‘2’ = one lesion less than 5 mm in diameter; ‘3’ = several lesions or less than 25% of the grape berry surface infected; and ‘4’ = more than 25% of the grape berry surface infected, and if sporulation was observed. The decay index (DI) was calculated using the following formula and expressed as % [29]:(1)$$\mathrm{DI}\;=\;\frac{\sum \left(d\times f\right)}{N\times D}$$
where d is the degree of rot severity scored on a grape berry and f is its respective quantity, N is the total number of grape berries examined, and D is the highest degree of disease severity occurring on the scale shown above.

### 2.5. Stem Browning Assessment

The browning grade was determined by visual scoring on a 0–4 scale according to the color change of the grape stem [30]. The scoring criteria use was as follows: ‘0’ meant the cap stem was green and healthy; ‘1’ represents a slight browning of the cap stem; ‘2’ indicates a slight to moderate browning of the cap stem; ‘3’ represents a moderate browning of the cap stem and secondary stem; and ‘4’ meant the cap stem and secondary stem are fully brown. When no browning was observed, the stem browning grade was assigned a zero; when there was full browning, the grade of stem browning was four. The grape berries were evaluated in duplicate by a team of individuals with a central person confirming the agreement in stem browning assessment and expressed as %.

### 2.6. Chemical Analysis of Grape Juice: Measurement of Total SSC, pH, and TA

Grape juice was obtained using a food blender after the seeds were removed. The juice was then filtered through four layers of cheesecloth in preparation for measurement of total SSC and pH. Total SSC was measured by a hand-held refractometer (RHB-18ATC, Shanghai, China) at 20 °C and the appropriate temperature correction adjustments were applied, and the results were expressed as %. The pH was determined by using a PB-10 pH meter (Sartorius, Göttingen, Germany). TA was determined by potentiometric titration with 0.1 N NaOH up to pH = 8.2 and expressed in % of tartaric acid [31]. All experiments were performed in quadruplicate.

### 2.7. Assay of Total Phenols

Total phenols were measured according to the method described by Sánchez-González et al. [16]. Table grape tissue (35 g) was suspended in 40 mL of methanol and 10 mL of 6 N HCl. The mixture was homogenized in a grinder (Jiuyang, Jinan, China) for 5 min, and then the supernatant was obtained by centrifugation at 10,000× *g* for 10 min at 4 °C. The total phenol content in the table grape berry tissue supernatant was determined based on the Folin–Ciocalteu method [16]. The supernatant (250 μL), ultrapure water (15 mL), and 1.25 mL of the Folin–Ciocalteu reagent (Solarbio, Beijing, China) were added in a 25 mL brown volumetric flask and held for 8 min. Then, 7.5% Na_2_CO_3_ (3.75 mL) was added followed by ultrapure water to a total volume of 25 mL. The reaction mixture in the brown volumetric flask was incubated in darkness for 2 h at room temperature. The absorbance of the reaction mixture was measured using a U-2910 spectrophotometer at 765 nm. A gallic acid standard (Beijing Solarbio Science & Technology Co., Ltd., Beijing, China) was used to generate a standard curve, and the total phenols were expressed as mg of gallic acid equivalents per gram of grape berry weight. The absorbance measurement of each sample was carried out in triplicate.

### 2.8. Microbiological Analysis

Aerobic mesophilic bacteria (AMB), yeast and molds (YAMs), and coliform bacteria (CB) counts were evaluated in all groups throughout the entire grape storage period. Sterile physiological saline (225 mL) was added to each sample and control group (25 g of grape tissue/group). The microbes present on the surface of the table grapes were fully dissolved in the physiological saline with the aid of a constant temperature shaking incubator (SPH-2102, Shanghai Shiping Experimental Equipment Co., Ltd., Shanghai, China) at 60 rpm for 35 min at 25 °C. Serial dilutions of grape surface microbial fluid were plated on plate count agar (PCA) (Qingdao Hope Bio-Technology Co., Ltd., Qingdao, China) to determine AMB, and potato dextrose agar (PDA) (Beijing Aoboxing Bio-Technology Co., Ltd., Beijing, China) and violet red bile agar (VRBA, Qingdao Hope Bio-Technology Co., Ltd., Qingdao, China) were used for enumeration of AMB, YAMs, and CB. PCA, PDA, and VRBA plates were incubated for 48 h at 30 °C, for 5 d at 25 °C, and for 1–2 d at 37 °C, respectively. Microbial counts were evaluated at 0, 5, 10, 15, and 20 d of sample storage at 25 °C. The results were expressed as log CFU/g grape berry weight, and each treatment of four replicates was analyzed in duplicate.

### 2.9. Sensory Evaluation

Sensory evaluation of table grapes was based on the observed glossiness, color, odor, appearance, and overall acceptability parameters [14,32], and the tasting parameters were not required for this step. The sensory quality, purchase intention, and acceptable evaluation test were performed by the panelists on day 0, 5, 10, 15, and 20. Panelists were individuals that frequently purchased and consumed grapes. Purchase intention was assessed with a five-point structured hedonic scale, ranging from 1 to 5: a ‘1’ meant they certainly would not buy the grapes, and a ‘5’ meant they certainly would buy the grapes. In addition, a nine-point hedonic scale was used with the following scoring definitions: a ‘1’ meant an extreme dislike for the grapes, a ‘5’ meant they neither had a like nor dislike for the grapes, and a ‘9’ meant they had an extreme liking for the grapes [33]. The sensory evaluation of the grape berries was evaluated in duplicate by a team of seven individuals with a central person confirming the agreement in sensory assessment.

### 2.10. Statistical Analysis

The results of physicochemical properties and microbial counts were expressed as the means ± standard deviation of four parallel samples. The statistical significance of the difference between the three groups was tested using analysis of variance (ANOVA). The Tukey test was applied for post hoc comparisons. Pearson correlation analysis was performed by selecting indicators to evaluate the extent of relationships between the groups. All of the above statistical analyses were completed using IBM SPSS Statistical Analysis Software v. 22.0.0.0 (SPSS Inc., Somers, NY, USA) for Windows software, and the significance level was defined at *p* < 0.05.

## 3. Results

### 3.1. Weight Loss

Figure 1 shows the weight loss of the control and the *Lactobacillus* and *Leuconostoc* supernatant-treated grape berries throughout the storage period. Weight loss was significantly higher in the control grape berries than in the grape berries treated with supernatants from strains F17 and H52 from day 10 to 20 (*p* < 0.05).

### 3.2. The Effects on Decay of Fruit

The results of the control and the *Lactobacillus* and *Leuconostoc* supernatant-treated grape berries from 0 to 20 d is depicted in Figure 2. The decay of untreated grape berries began by day 5 of storage and was greater than that of the *Lactobacillus* and *Leuconostoc* supernatant-treated berries. In addition, the reduction of decay was more pronounced for the F17 supernatant-treated grape berries, which had the lowest decay rate throughout the experiment.

### 3.3. Grade of Stem Browning

Figure 3 shows that stem browning occurred most often in the untreated grapes, and *Lactobacillus* and *Leuconostoc* supernatant-treated grapes when the storage time was extended. The control group had almost complete stem browning by the end of the storage period, and the *Lactobacillus* and *Leuconostoc* supernatant-treated grapes showed significant differences at 5, 10, and 20 d (*p* < 0.05) in comparison with the control grapes.

### 3.4. Chemical Analysis of Grape Juice

Changes in SSC, TA, SSC/TA, and pH values of grapes in the control, *Lactobacillus*, and *Leuconostoc* supernatant-treated groups are listed in Table 1. Total SSC of *Lactobacillus* and *Leuconostoc* supernatant-treated grapes increased gradually with increased storage time and the maximum value was reached at the end of storage, whereas the control grapes peaked at day 10 of storage and then progressively decreased. The total SSC of the strain H52 supernatant-treated grapes increased throughout storage, but decreased below the total SSC of the grapes treated with the strain H52 supernatant on day 20. The grape berries treated with the supernatant from strain F17 maintained a high level of total SSC throughout the storage period.

The TA of grapes in all groups gradually decreased over the entire storage period, while the largest TA decrease was observed in the control grapes, from 0.67 to 0.5, at Δ = 0.17 (Table 1). In general, the rate of SSC/TA in all samples showed an upward trend with storage time and reached a maximum value at 20 d of storage. The value of SSC/TA in the *Lactobacillus* and *Leuconostoc* supernatant-treated grapes progressively increased and reached a maximum value at 20 d, while the value of SSC/TA in the control grapes increased through day 10 to 32.52 and then decreased at day 15, and slightly increased again at day 20. The SSC/TA from the strain F17 supernatant-treated grapes was always a somewhat lower value than the SSC/TA of the strain H52 supernatant-treated grapes with the SSC/TA at day 20 of F17 and H52 achieving 36.19 and 36.83, respectively. Table 1 shows the pH of grapes in all samples increased with storage time and reached a maximum value by the end of the storage period. However, the pH values of the strains F17 and H52 supernatant-treated grapes showed the lowest values following the 5d storage period. Overall, the *Lactobacillus* and *Leuconostoc* supernatant-treated grapes showed the lowest pH values in comparison with the control grapes at all storage times.

### 3.5. Changes in Total Phenols

The effects of different treatments on total phenols are shown in Figure 4. During the storage period, total phenols decreased in all groups over the storage time. Total phenols of the *Lactobacillus* and *Leuconostoc* supernatant-treated grapes were significantly higher than that of the control grapes at all storage times (*p* < 0.05), and the *Lactobacillus* strain F17 supernatant-treated grapes had the highest total phenols.

### 3.6. Microbiological Analysis

The results of AMB analysis for the untreated, *Lactobacillus*, and *Leuconostoc* supernatant-treated grapes are shown in Table 2. Before treatments, the initial counts of AMB were about 2.14 log CFU/g. In addition, the changes in the AMB counts for all groups increased from day 5 to day 15, and the counts reached their maximum value at day 15. The number of AMB counts in the F17 and H52 supernatant-treated grapes were lower than that in the control grapes when the storage period was 10, 15, and 20 d. F17 supernatant-treated grapes showed a significant difference between the control and the strain H52 supernatant-treated grapes after storage at 10 and 15 d (*p* < 0.05).

The changes in the number of YAMs in all samples are presented in Table 2. At 0 d, the initial value of YAMs was about 1.44 log CFU/g, but with increased storage time, the number of YAMs in all samples increased. The number of YAMs in the *Lactobacillus* strain F17 and *Leuconostoc* strain H52 supernatant-treated grapes resulted in lower counts in comparison with the control grapes, and the *Leuconostoc* strain H52 supernatant-treated grapes showed still lower YAMs values with a significant difference compared with the other two groups at 10, 15, and 20 d storage periods (*p* < 0.05).

The initial population of CB at time 0 on the table grapes was about 2.28 log CFU/g, as shown in Table 2. The CB counts of all supernatant-treated table grapes were lower than the control grapes, and *Leuconostoc* strain H52 supernatant-treated grapes had the lowest number of CB counts among all groups during the entire storage period (*p* < 0.05). The number of CB from the strain H52 supernatant-treated table grapes was about 2.51 log CFU/g at 20 d; it was reduced by 0.27 log CFU/g in comparison with the control grapes. It should also be noted that the supernatant of strain F17 significantly inhibited CB during the storage periods of 5, 15, and 20 d (*p* < 0.05).

### 3.7. Sensory Evaluation

The sensory evaluation scores of all samples decreased with increasing storage time, and all samples reached the lowest value at 20 days (Figure 5). The sensory evaluation score of the *Lactobacillus* and *Leuconostoc* supernatant-treated table grapes was higher than the untreated table grapes. The strain F17 supernatant-treated table grapes had the highest sensory scores throughout postharvest storage and were significantly different than the other two groups.

### 3.8. Pearson Correlation Analysis

The Pearson correlation analysis was used to verify the correlations between the experimental indicators, such as weight loss (WL), decay rate (DR), stem browning (SB), pH, total soluble solids content (SSC), titratable acidity (TA), total phenols (TP), microbial counts, and sensory evaluation (SE), and only the indicators of significant correlation are listed in Table 3. The weight loss of table grapes showed a significant positive correlation with decay rate, stem browning, and total SSC. Moreover, weight loss of table grapes showed a significant negative correlation with the sensory evaluation. Likewise, the decay rate of table grapes had a significant positive correlation with stem browning and a negative correlation with sensory evaluation and total phenols. Additionally, pH showed a significant negative correlation with TA and total phenols, and a negative correlation was observed between stem browning and sensory evaluation. AMB, YAMs, and CB values exhibited a positive correlation with weight loss and decay rate, while they showed a significant negative correlation with sensory evaluation.

## 4. Discussion

### 4.1. Weight Loss

The grape berry skin acts as a barrier to protect against water loss and invasion of fungal pathogens, and it contributes to the control of gas exchange [34]. Respiration and transpiration of postharvest grape berries were the main factors causing weight loss, and migration of water from the berry to the environment causes berry weight loss during storage [35]. In addition, weight loss may also be partly attributed to increased metabolic activity of grapes [16]. It is worth noting that too much weight loss may cause deterioration of the grape berries’ appearance and even affect their sensory appeal and marketability [36]. In our studies, grape berry treatment by the supernatants of strains F17 and H52 significantly reduced the weight loss of table grapes, which indicates that *Lactobacillus* and *Leuconostoc* supernatant-treatment had a positive effect on maintaining the sensory quality of grapes. As previously shown in a study with strawberry fruit, *Lactobacillus* intact cells and cell lysates adhered naturally to the surface of the berries, resulting in a biofilm formation and enhanced water preservation [37].

### 4.2. Effects on the Decay of Fruit

Decay rate was a key parameter for evaluation of commercial quality and postharvest shelf-life of grape berries. It has been reported that gray mold is considered the main pathogen causing postharvest decay of table grapes [38]. The *Lactobacillus* supernatant-treated grape berries had a lower decay rate compared with the control berries, which may be a result of having products in the bacterial supernatants. In this regard, Dalié et al. [39] reported that *Lactobacillus* produces antifungal low molecular weight compounds. Another possible reason for the observed protection may be the micro-acidic environment formed by the *Lactobacillus* supernatant-treatment.

### 4.3. Grade of Stem Browning

The grade of stem browning not only reflects the freshness of grapes, but also has a direct effect on the consumer’s intention to purchase the grapes. Moreover, browning and drying of stems may begin to occur even before the actual deterioration of fruit [40]. Thus, stem browning is often used to evaluate the quality of table grapes from a sensory perspective [9]. In these studies, the *Lactobacillus* and *Leuconostoc* supernatant-treated grapes showed lower stem browning than the untreated table grapes during the entire storage period. There are two possible reasons for this result. Desiccation is considered the main factor for stem browning, and weight loss of the untreated grapes in this study was significantly higher than that of *Lactobacillus* or *Leuconostoc* supernatant-treated grapes. The supernatants of strains F17 and H52 may have possessed favorable antioxidant capacity, which may inhibit the activity of oxidase, resulting in reduced stem browning.

### 4.4. Chemical Analysis of Grape Juice

Total SSC, TA, SSC/TA, and pH are key indicators reflecting the organoleptic quality, maturity, and senescence of table grapes [9,41]. Total SSC of the *Lactobacillus* and *Leuconostoc* supernatant-treated grapes increased gradually over the length of storage, and reached a maximum value by the end of storage. This result was consistent with the report by Meng et al. [28] showing that the total SSC of postharvest table grapes gradually increased with the maturity of the grapes stored at 20 °C. The decrease in total SSC observed in the control grapes at 15 to 20 d may be explained by the large loss of water at these times, which is consistent with the weight loss data. The grapes treated with the strain H52 supernatant maintained a high level of total SSC throughout the storage period, and this result may be related to the secretion of exopolysaccharide by strain H52.

Organic acids, primarily tartaric acid, play a key role in grape organoleptic quality. Moreover, the change of the tartaric acid content is one of the indicators used to measure the grape’s physiological processes and fresh-keeping ability [42]. The TA content is used as a marker to determine when the respiratory substances of postharvest grapes are consumed, and *Lactobacillus* and *Leuconostoc* supernatant-treatments appear to reduce the table grape’s respiration. The reduction in TA correlates with the loss of water in the control grapes. The SSC/TA ratio value is related to the maturity of grape berries and may be used as an indicator to determine the postharvest senescence of table grapes [43]. We observed a progressive increase of SSC/TA in the *Lactobacillus* and *Leuconostoc* supernatant-treated grapes, which reached maximum values at day 20, while the value of the SSC/TA ratio in the control grapes increased only until day 10 and then decreased by day 15. These observations may be interpreted as the supernatant-treatments can delay the onset of maturity and senescence of table grapes. pH has been used as an indicator that reflects the senescence of grapes [44]. It was shown that, with the prolongation of storage time, the pH increased owing to utilization of the grape’s organic acids during respiration [45]. However, pH values of the strains F17 and H52 supernatant-treated grapes presented the lowest values after being stored for 5 d, and a plausible reason may be that strains F17 and H52 produce acidic metabolic products such as lactic acid [20,21], or the *Lactobacillus* and *Leuconostoc* supernatant-treatments may suppress the decomposition of organic acids by reducing the respiration of grapes.

### 4.5. Changes in Total Phenols

Polyphenol is an important indicator used to evaluate the internal postharvest quality of grapes, which reflects the antioxidant degree and quality of grapes [46]. The change in total phenols is closely related to the degree of browning. Polyphenols will consume dissolved oxygen and, together with the action of polyphenol oxidase, will oxidize and polymerize to form quinone substances, which then undergo enzymatic browning [16]. In these studies, the *Lactobacillus* and *Leuconostoc* supernatant-treatments significantly increased total phenol production (*p* < 0.05) and the strain F17 supernatant-treated grapes exhibited the greatest retention of total phenols. A possible explanation for this result is that strains F17 and H52 possessed a favorable antioxidative capacity and had an antioxidant effect on the grapes.

### 4.6. Microbiological Analysis

The AMB counts for all groups increased from day 5 to day 15, but the AMB counts of the *Lactobacillus* and *Leuconostoc* supernatant-treated groups were lower than that in the control group when the storage period was 10, 15, and 20 d. Additionally, AMB counts showed a significant difference in all groups at a storage period of 10 and 15 d (*p* < 0.05). A likely reason for these observations is that the products of *Lactobacillus* and *Leuconostoc*, such as bacteriocins, extracellular polysaccharides, and lactic acid, inhibit aerobic mesophilic bacteria; however, inhibition mechanisms are unclear.

The number of YAMs in the *Lactobacillus* and *Leuconostoc* supernatant-treated groups showed a lower number of YAMs counts in comparison with the control group, and can be explained by the production of antifungal low-molecular-weight-compounds, such as organic acids, reuterin, hydrogen peroxide, proteinaceous compounds, hydroxyl fatty acids, and phenolic compounds [39]. The strain H52 supernatant-treatment significantly inhibited YAMs among all groups from day 10 to 20 and may be a result of production of an extracellular polysaccharide [47].

It was reported that bacterial exopolysaccharides produced by *Lactobacillus* showed significant antibacterial activity against pathogens such as *Cronobacter sakazakii*, *Escherichia coli*, *Listeria monocytogenes*, *Salmonella* Typhimurium, *Shigella sonnei,* and *Staphylococcus aureus* [48]. It should also be noted that the strain F17 supernatant-treatment also significantly inhibited CB during the storage periods of 5, 15, and 20 d (*p* < 0.05). Inhibition of CB may be a result of a bacteriocin, a product of strain F17 [18].

### 4.7. Sensory Evaluation

The sensory evaluation score of table grapes was based on the comprehensive evaluation of the glossiness, color, odor, overall appearance, and acceptability parameters, and reflects the consumers’ acceptance of postharvest table grapes. The strain F17 supernatant-treated table grapes had the highest sensory scores and were significantly different among the three groups. This demonstrated that the strain F17 supernatant-treatment significantly improved the sensory quality of postharvest table grapes. Therefore, from a business standpoint, the strain F17 supernatant-treatment was more suitable for table grape preservation.

The weight loss of table grapes showed a significant positive correlation with decay rate, stem browning, and total SSC, and was consistent with the results of Champa et al. [49], who reported that weight loss of the grape berry was positively correlated with decay rate. Previous studies have shown that even a minimal moisture loss could cause visible quality changes in table grapes, such as browning [30], and wilting and desiccation [50]. With increased storage time, progressive increases in total SSC may be interpreted as water evaporation from the berry surface [51]. Weight loss of table grapes also showed a significant negative correlation with the sensory evaluation [49], and water loss would lead to deterioration of grape berries. Likewise, Champa et al. [49] found that the decay rate of table grapes had a significant positive correlation with stem browning and a negative correlation with sensory evaluation and total phenols. In our study, grape decay may lead to the loss of polyphenols. With increased storage time, TA was gradually consumed by grape respiration [52], resulting in the increase of pH and stem browning [45], which decreased the sensory evaluation scores [53]. The AMB, YAMs, and CB exhibited a positive correlation with weight loss and decay rate of the grape berries, while showing a significant negative correlation with sensory evaluation. These results are consistent because microbes consume nutrients and produce harmful substances that can cause grape weight loss and increase the decay rate, which will increase a negative sensory evaluation of grapes. These results suggest that weight loss, decay rate, stem browning, total SSC and microorganisms are highly correlated with the sensory evaluation and quality of postharvest grapes. Therefore, the development of a biopreservative is expected to have a positive effect on these indicators.

## 5. Conclusions

In this study, *Lactobacillus delbrueckii* subsp. *bulgaricus strain* F17 and *Leuconostoc lactis* strain H52 supernatants possessing probiotic properties were developed as potential biopreservatives for postharvest table grape preservation. Although, supernatants from both strains F17 and H52 significantly reduced weight loss, decay rate, and delayed maturity and senescence of table grapes, the supernatant from strain F17 was more effective and resulted in better grape sensory evaluations. For YAMs, strain F17 demonstrated significant initial inhibition in the first five days, while the supernatant of strain H52 showed a significantly greater growth inhibition during the entire storage period. In addition, the Pearson correlation analysis results suggested that weight loss, decay rate, stem browning, total SSC, and microorganisms were all highly correlated with the sensory evaluation and quality of postharvest grapes. On the basis of the overall characteristics and sensory quality of the treated grapes, the supernatant containing products of strain F17 was the overall best potential biopreservative for improving the postharvest quality of ‘Red Globe’ grapes.

## Figures and Tables

**Figure 1 microorganisms-08-00656-f001:**
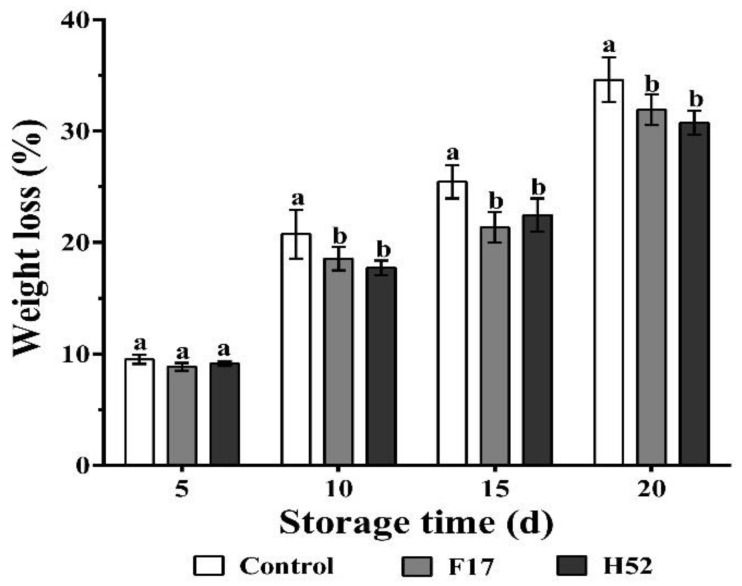
Changes in weight loss with different treatments during storage at 25 °C. Different lowercase letters at the same storage time indicate a significant difference (*p* < 0.05).

**Figure 2 microorganisms-08-00656-f002:**
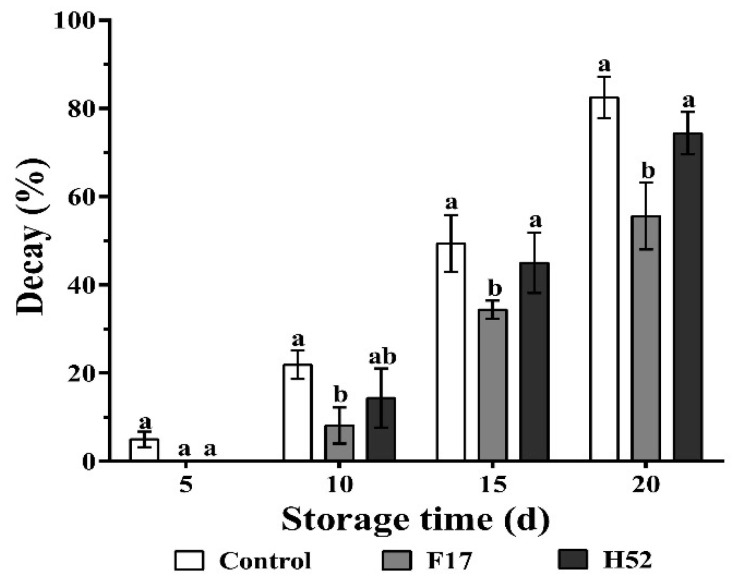
Changes in decay percentage with different treatments during storage at 25 °C. Different lowercase letters at the same storage time indicate a significant difference (*p* < 0.05).

**Figure 3 microorganisms-08-00656-f003:**
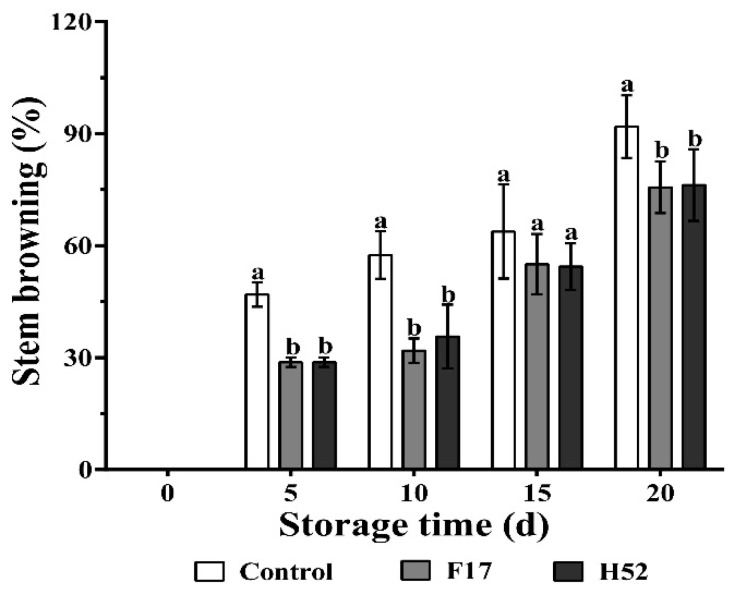
Changes in stem browning with different treatments during storage at 25 °C. Different lowercase letters at the same storage time indicate a significant difference (*p* < 0.05).

**Figure 4 microorganisms-08-00656-f004:**
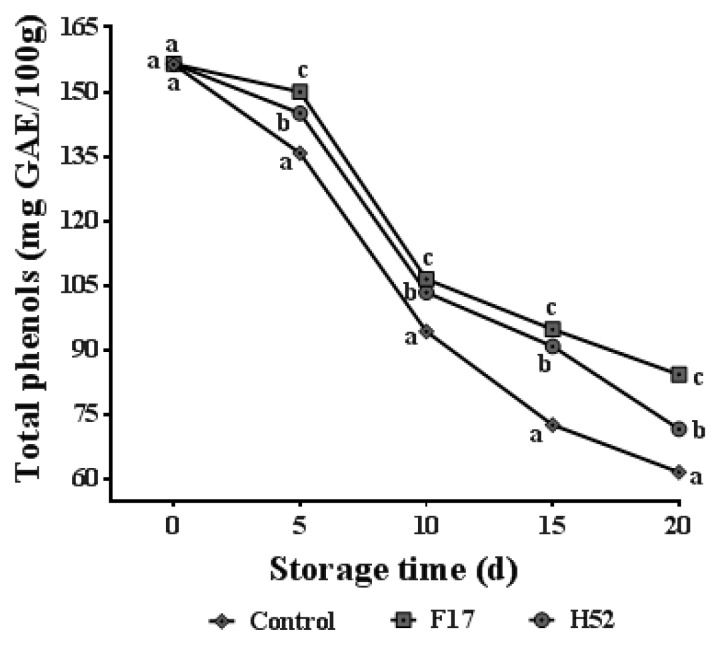
Changes in total phenols of table grapes with treatments during storage at 25 °C. Each point is the average and standard deviation of four replicates. Different lowercase letters at the same storage time indicate a significant difference (*p* < 0.05).

**Figure 5 microorganisms-08-00656-f005:**
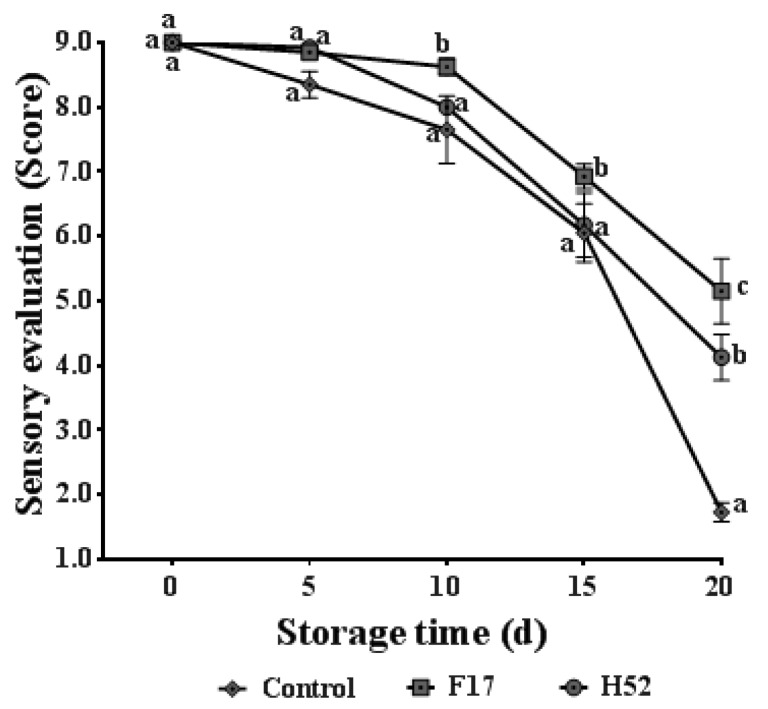
Changes in sensory evaluation of table grapes with different treatments during storage at 25 °C. Each point is the average and standard deviation of four replicates. Different lowercase letters at the same storage time indicate a significant difference (*p* < 0.05).

**Table 1 microorganisms-08-00656-t001:** SSC (%), TA (%), SSC/TA, and pH with different treatments on ‘Red Globe’ table grapes during storage at 25 °C.

Indicators	Treatment	Storage Times (Days)				
		0	5	10	15	20
SSC	Control	15.00 ± 0.10 ^a^	16.00 ± 0.71 ^a^	19.63 ± 0.65 ^a^	18.00 ± 0.61 ^a^	17.45 ± 0.36 ^a^
	F17	15.00 ± 0.10 ^a^	15.38 ± 0.41 ^a^	17.50 ± 0.35 ^b^	18.38 ± 1.08 ^a^	20.25 ± 0.25 ^b^
	H52	15.00 ± 0.10 ^a^	15.75 ± 0.56 ^a^	18.25 ± 0.25 ^b^	19.88 ± 0.74 ^b^	19.50 ± 0.50 ^b^
TA	Control	0.67 ± 0.02 ^a^	0.64 ± 0.01 ^a^	0.60 ± 0.01 ^a^	0.57 ± 0.02 ^a^	0.50 ± 0.01 ^a^
	F17	0.68 ± 0.02 ^a^	0.67 ± 0.01 ^a^	0.63 ± 0.01 ^a^	0.62 ± 0.03 ^b^	0.56 ± 0.02 ^b^
	H52	0.68 ± 0.03 ^a^	0.65 ± 0.02 ^a^	0.61 ± 0.01 ^a^	0.59 ± 0.00 ^ab^	0.53 ± 0.01 ^ab^
SSC/TA	Control	22.49 ± 0.65 ^a^	25.01 ± 1.33 ^a^	32.52 ± 1.45 ^a^	31.90 ± 1.67 ^ab^	34.73 ± 0.56 ^a^
	F17	22.13 ± 0.86 ^a^	23.04 ± 0.54 ^a^	27.89 ± 0.67 ^b^	29.80 ± 1.85 ^b^	36.19 ± 1.13 ^a^
	H52	22.27 ± 1.12 ^a^	24.42 ± 0.61 ^a^	30.08 ± 1.16 ^ab^	33.59 ± 1.91 ^a^	36.83 ± 1.52 ^a^
pH	Control	3.51 ± 0.02 ^a^	3.52 ± 0.03 ^a^	3.54 ± 0.02 ^a^	3.57 ± 0.01 ^a^	3.63 ± 0.02 ^a^
	F17	3.48 ± 0.02 ^a^	3.45 ± 0.02 ^b^	3.49 ± 0.03 ^a^	3.52 ± 0.02 ^b^	3.56 ± 0.05 ^b^
	H52	3.48 ± 0.02 ^a^	3.46 ± 0.03 ^b^	3.51 ± 0.01 ^a^	3.55 ± 0.02 ^ab^	3.60 ± 0.02 ^ab^

SSC = soluble solids content; TA = titratable acidity; data = mean ± standard deviation (SD) of three replicates. ^a,b^ Different lowercase superscripts indicate a significant difference (*p* < 0.05).

**Table 2 microorganisms-08-00656-t002:** Changes in microbial loads (log colony forming units (CFU)/g) of aerobic mesophilic bacteria (AMB), yeast and molds (YAMs), and coliform bacteria (CB) with different bacterial supernatant-treatments on table grapes during storage at 25 °C.

Microorganism	Treatment	Storage Times (Days)				
		0	5	10	15	20
AMB	Control	2.14 ± 0.10 ^a^	2.30 ± 0.03 ^a^	3.14 ± 0.34 ^a^	3.15 ± 0.04 ^a^	2.51 ± 0.12 ^a^
	F17	2.12 ± 0.19 ^a^	2.30 ± 0.00 ^a^	2.51 ± 0.13 ^b^	2.61 ± 0.08 ^b^	2.47 ± 0.06 ^a^
	H52	2.14 ± 0.09 ^a^	2.30 ± 0.05 ^a^	2.85 ± 0.17 ^c^	2.91 ± 0.15 ^c^	2.35 ± 0.04 ^a^
YAMs	Control	1.44 ± 0.07 ^a^	2.55 ± 0.16 ^a^	3.48 ± 0.15 ^a^	3.41 ± 0.11 ^a^	3.58 ± 0.16 ^a^
	F17	1.45 ± 0.04 ^a^	1.80 ± 0.05 ^b^	3.33 ± 0.18 ^a^	3.25 ± 0.16 ^a^	3.54 ± 0.04 ^a^
	H52	1.44 ± 0.02 ^a^	1.89 ± 0.07 ^b^	2.26 ± 0.11 ^b^	2.99 ± 0.15 ^b^	3.10 ± 0.06 ^b^
CB	Control	2.28 ± 0.06 ^a^	2.56 ± 0.11 ^a^	2.47 ± 0.04 ^a^	3.22 ± 0.07 ^a^	2.78 ± 0.07 ^a^
	F17	2.24 ± 0.08 ^a^	2.11 ± 0.07 ^b^	2.35 ± 0.02 ^a^	3.05 ± 0.04 ^b^	2.59 ± 0.06 ^b^
	H52	2.28 ± 0.02 ^a^	2.00 ± 0.08 ^b^	1.97 ± 0.09 ^b^	2.70 ± 0.09 ^c^	2.51 ± 0.10 ^b^

^a,b,c^ Different lowercase superscripts indicate significant differences (*p* < 0.05).

**Table 3 microorganisms-08-00656-t003:** Correlation between some selected quality parameters of table grapes treated and untreated with strains F17 or H52 supernatants during 20 d of 25 °C storage.

Variables Compared	Pearson Correlation Coefficient (*r*)		
	Control Group	F17 Group	H52 Group
WL vs. DR	0.930 **	0.884 **	0.902 **
WL vs. SSC	0.768 **	0.938 **	0.953 **
WL vs. SB	0.926 **	0.941 **	0.957 **
WL vs. SE	−0.879 **	−0.862 **	−0.910 **
WL vs. AMB	0.572 **	0.767 **	0.463 *
WL vs. YAMs	0.907 **	0.933 **	0.956 **
WL vs. CB	0.647 **	0.600 **	0.474 *
DR vs. pH	0.885 **	0.744 **	0.931 **
DR vs. SB	0.832 **	0.878 **	0.887 **
DR vs. SE	−0.981 **	−0.959 **	−0.970 **
DR vs. TP	−0.910 **	−0.885 **	−0.904 **
DR vs. AMB	0.627 **	0.756 **	0.568 **
DR vs. YAMs	0.829 **	0.843 **	0.908 **
DR vs. CB	0.763 **	0.852 **	0.620 **
pH vs. TA	−0.821 **	−0.457 *	−0.851 **
pH vs. TP	−0.815 **	−0.688 **	−0.865 **
SE vs. SSC	−0.489 *	−0.899 **	−0.949 **
SE vs. SB	−0.826 **	−0.894 **	−0.884 **
SE vs. AMB	−0.613 **	−0.721 **	−0.594 **
SE vs. YAMs	−0.823 **	−0.866 **	−0.926 **
SE vs. CB	−0.489 *	−0.616 **	−0.693 *

* The value was significant at *p* < 0.05; ** the value was significant at *p* < 0.01. WL = weight loss; DR = decay rate; SB = stem browning; TP = total phenols; SE = sensory evaluation.

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
