# Peer review of "Products of Lactobacillus delbrueckii subsp. bulgaricus Strain F17 and Leuconostoc lactis Strain H52 Are Biopreservatives for Improving Postharvest Quality of ‘Red Globe’ Grapes"

_microorganisms, 2020, doi:10.3390/microorganisms8050656_

Round 1
Reviewer 1 Report
The aim of this work was to determine whether the secondary metabolites of Lactobacillus delbrueckii subsp. bulgaricus strain F17 and Leuconostoc lactics strain H52 could maintain post harvest quality sensors of “Red globe”table grapes and, if so, to improve quality and shelf life during storage. Samples treated and untreated were analyzed for physicochemical (weight loss, decay rate, stem browning),biochemical (pH, total solid content, titratable acidity and total phenols), microbiological properties and sensory evaluation for 20 days. Overall, both strains significantly reduced weight loss, decay rate, aerobic mesophilic bacteria and coliform bacteria counts, delayed maturity and senescence of table grapes and reduced titratable acidity and total phenols. However, the supernatant of strain F17 was more effective and resulted in better sensory evaluations and had a significant inhibitory effect on fungi over the whole storage periods suggesting a potential use of Lactobacillus delbrueckii subsp. bulgaricus as a preservative agent for table grape preservation.
Overall, the manuscript is well-written and structured. The study has been done carefully and the methods are appropriate. The discussion section is adequate. The paper is scientifically sound.In general, I find the manuscript very clearly written. Nevertheless, some sections like abstract and introduction, are quite long. English is fairly good and the text flow well.
Some specific comments
Title
page 1- lines 2-4: it must be changed by adding Leuconostoc lactics strain H52
Abstract
-lines 19-20: please delete ”and their supernatants were used to improve preservation of table grapes”.
- line 21:please delete ”were stored at 25 °C for 0, 5, 10, 15 and 20 d and”
- line 23: please add “for 20 days” after round bracket
- line 24: please delete ”F17 and H52”
- lines 28-29: please delete “but the colony counts on day 5 to 20 were only slightly less than that of the control group”
- lines 29-30: please change “yeast and molds” to “fungi”
Introduction
page 2
- line 55:please delete “once harvested do not ripen further”
- line 82:please delete “rather than using chemical preservatives”
- line 85:please write in capital letter the first letter of words “generally recognized as safe”
- line 91: please change “bacteria” in “bacterium”
- lines 93-95: please delete sentences ”that are often linked to outbreaks involving fresh produce caused by consumption of contaminated food .It is of great importance that strains F17 or H52 may be utilized to develop a biological preservative to control foodborne pathogens on table grapes.“ They are redundant.
page 3
- lines 96-98: please delete “Although Lactobacillus and their metabolites are widely used to control foodborne pathogens 96 and spoilage organisms in meat products to ensure food quality and safety, to the best of our knowledge”
- lines 105-106: please delete “and the strain having the best potential for improving the quality of postharvest table grapes was determined”
Materials and Methods
page 3
-lines 123-124: please delete “This process was repeated to be sure the bacteria were in the linear phase and”
- line 126: please delete “adjusting the bacterial concentrations” and add “that” after “after”
- line 141: please add “and expressed as %” before colon
page 4
- line 154: please add “and expressed as %”
- line 155: please change subtitle to Chemical analysis of grape juice: measurement of total SSC, pH and TA
- line 183: please change “bacteria fluid” to “microbial fluid”
- line 185: please change “bacterial count” to “AMB”
Results
page 5
- line 213: please add “and Leuconostoc” after “Lactobacillus”
Figure 1
please move “time 0” to the point of the axes intersection
page 6
- line 223: please add “and Leuconostoc” after “Lactobacillus”
Figure 2
please move “time 0” to the point of the axes intersection
- lines 233-235: please delete “The Lactobacillus and Leuconostoc supernatant-treated grapes showed less stem browning than the untreated table grapes over the entire storage period”
page 7
- line 245: please delete “the total SSC of “
- lines 246:please change F17 to H52
- line 248: please change H52 to F17... in these sentences the two strains were switched. It is the opposite of what is written.
- line 260:please add “and Leuconostoc” after “Lactobacillus”
page 8
- lines 269-271: “The Leuconostoc strain H52 supernatant-treated grapes significantly decreased in a downward trend with the control (P < 0.05), however, the Lactobacillus strain F17 supernatant-treated grapes had the highest total phenols.” This sentence is too complicates and difficult to understand. Please reformulate.
- lines 278,285,290: please add “about” before CFU/ml values(i.e. “were about 2.14 CFU/ml” )
- lines280-281: please change “Lactobacillus strain F17” to “F17 and H52”
- line282:please add a full stop after “20” and delete “and strain” before F17
- line 286: please add “and Leuconostoc strain H52” after “Lactobacillus strain F17”
- lines 288-289: please change “a significant difference among all three groups during “ to “ a significant difference compared to other two groups at”
- lines 290- 291:please delete “before the treatments were administered”
- line 291:please change “Lactobacillus” to “all”
- line 292: please delete “throughout the study”
- lines 295-296:please delete “the end of storage time, while the number of CB on the H52 supernatant-treated grapes were” and add “20 d: it was” after “2.51 log CFU/g at “
Discussion
page 11
-lines 364-368: complicated sentence. Please re-word. Preferably, split into two separate sentences.
page12
- lines 383-387: Actually, the pH also increased in control group. How do you explain it?
- line 396: please change “inhibited the downward trend of “ to “increased”
- lines 401-404: complicated sentence. Please re-word into two separate sentences.
- lines 412-413: “The strain H52 supernatant-treatment significantly inhibited YAMs among all groups over the entire storage period and may be a result of production of an extracellular polysaccharide”. Actually, YAMs decreased only between 10 to 20 days...
- lines 426-428: please delete “The obtained result also was consistent with the strongest purchase intention assessment. Additionally, table grapes treated with the strain H52 supernatant received an evaluation for purchase intention as 'liked moderately'.”
Conclusions
page13
- line 456:please delete “of YAMs”. It's a repetition
- lines 456-457:please delete “the supernatant of”
Author Response
The aim of this work was to determine whether the secondary metabolites of Lactobacillus delbrueckii subsp. bulgaricus strain F17 and Leuconostoc lactics strain H52 could maintain post harvest quality sensors of “Red globe”table grapes and, if so, to improve quality and shelf life during storage. Samples treated and untreated were analyzed for physicochemical (weight loss, decay rate, stem browning),biochemical (pH, total solid content, titratable acidity and total phenols), microbiological properties and sensory evaluation for 20 days. Overall, both strains significantly reduced weight loss, decay rate, aerobic mesophilic bacteria and coliform bacteria counts, delayed maturity and senescence of table grapes and reduced titratable acidity and total phenols. However, the supernatant of strain F17 was more effective and resulted in better sensory evaluations and had a significant inhibitory effect on fungi over the whole storage periods suggesting a potential use of Lactobacillus delbrueckii subsp. bulgaricus as a preservative agent for table grape preservation.
Overall, the manuscript is well-written and structured. The study has been done carefully and the methods are appropriate. The discussion section is adequate. The paper is scientifically sound.In general, I find the manuscript very clearly written. Nevertheless, some sections like abstract and introduction, are quite long. English is fairly good and the text flow well.
Thank you for your comments concerning our manuscript entitled “Secondary metabolites of Lactobacillus delbrueckii subsp. bulgaricus strain F17 as a biopreservative for improving postharvest quality of 'Red Globe' grapes” (Manuscript ID: microorganisms-765723). Those comments are all valuable and very helpful for revising and improving our paper, as well as an important significant guide for our research. We have studied your comments carefully and have made corrections to our manuscript which we hope will meet with your approval. In the revised version, the red colored words were the responses to the comments. Please note that when line numbers are specified in our responses, these refer to the revised manuscript. We highly appreciate the constructive and kind comments that all the reviewers addressed on our manuscript.
The main corrections in the paper and the responds to the reviewer’s comments are the following:
Some specific comments
Title
page 1
Point 1: - lines 2-4: it must be changed by adding Leuconostoc lactis strain H52
Response 1: Thank you very much for your great suggestion. It has been corrected. Please see line 2.
Abstract
Point 2: -lines 19-20: please delete “and their supernatants were used to improve preservation of table grapes”.
Response 2: It has been deleted. Please see line 20, thank you.
Point 3: - line 21: please delete “were stored at 25°C for 0, 5, 10, 15 and 20 d and”
Response 3: It has been deleted. Please see lines 20, thank you.
Point 4: - line 23: please add “for 20 days” after round bracket
Response 4: It has been corrected. Please see lines 22-23, thank you.
Point 5: - line 24: please delete “F17 and H52”
Response 5: It has been deleted. Please see line 23, thank you.
Point 6: -line 28-29: please delete “but the colony counts on day 5 to 20 were only slightly less than that of the control group”
Response 6: It has been deleted. Please see lines 27, thank you.
Point 7: - lines 29-30: please change “yeast and molds” to “fungi”
Response 7: It has been corrected. Please see line 28, thank you.
Introduction
page 2
Point 8: - line 55: please delete “once harvested do not ripen further”
Response 8: It has been deleted. Please see line 53, thank you.
Point 9: - line 82: please delete “rather than using chemical preservatives”
Response 9: It has been deleted. Please see line 80, thank you.
Point 10: - line 85: please write in capital letter the first letter of words “generally recognized as safe”
Response 10: It has been corrected. Please see lines 82-83, thank you.
Point 11: - line 91: please change “bacteria” in “bacterium”
Response 11: It has been corrected. Please see line 88, thank you.
Point 12: - lines 93-95: please delete sentence “that are often linked to outbreaks involving fresh produce caused by consumption of contaminated food. It is of great importance that strains F17 or H52 may be utilized to develop a biological preservative to control foodborne pathogens on table grapes.” They are redundant.
Response 12: It has been deleted. Please see lines 91, thank you.
page 3
Point 13: - lines 96-98: please delete “Although Lactobacillus and their metabolites are widely used to control foodborne pathogens 96 and spoilage organisms in meat products to ensure food quality and safety, to the best of our knowledge”
Response 13: Thank you very much for your kind suggestion. After our careful discussion and consideration, we think this sentence links our research with the journal and highlight our research characteristics. Anyway, we thank for your thoughtful advice.
Point 14: - lines 105-106: please delete “and the strain having the best potential for improving the quality of postharvest table grapes was determined”
Response 14: It has been deleted. Please see line 101, thank you.
Materials and Methods
page 3
Point 15: -lines 123-124: please delete “This process was repeated to be sure the bacteria were in the linear phase and”
Response 15: It has been deleted. Please see line 118, thank you.
Point 16: - line 126: please delete “adjusting the bacterial concentrations” and add “that” after “after”
Response 16: It has been changed. Please see line 120, thank you.
Point 17: - line 141: please add “and expressed as %” before colon
Response 17: It has been changed. Please see lines 134-135, thank you.
page 4
Point 18: - line 154: please add “and expressed as %”
Response 18: It has been corrected. Please see line 148, thank you.
Point 19: - line 155: please change subtitle to Chemical analysis of grape juice: measurement of total SSC, pH and TA
Response 19: It has been corrected. Please see line 149, thank you.
Point 20: - line 183: please change “bacteria fluid” to “microbial fluid”
Response 20: It has been corrected. Please see line 177, thank you.
Point 21: - line 185: please change “bacterial count” to “AMB”
Response 21: It has been corrected. Please see line 178, thank you.
Results
page 5
Point 22: - line 213: please add “and Leuconostoc” after “Lactobacillus”
Response 22: It has been corrected. Please see line 207, thank you.
Point 23: Figure 1
please move “time 0” to the point of the axis intersection
Response 23: It has been corrected. Please see Figure 1 on the line 211 in the manuscript, thank you.
page 6
Point 24: - line 223: please add “and Leuconostoc” after “Lactobacillus”
Response 24: It has been corrected. Please see line 217, thank you.
Point 25: Figure 2
please move “time 0” to the point of the axis intersection
Response 25: It has been corrected. Please see Figure 2 on the line 220 in the manuscript, thank you.
Point 26: - lines 233-235: please delete “The Lactobacillus and Leuconostoc
supernatant-treated grapes showed less stem browning than the untreated table grapes over the entire storage period”
Response 26: It has been deleted. Please see lines 227, thank you.
page 7
Point 27: - line 245: please delete “the total SSC of “
Response 27: It has been deleted. Please see line 238, thank you.
Point 28: - lines 246: please change F17 to H52
Response 28: It has been corrected. Please see line 239, thank you.
Point 29: - line 248: please change H52 to F17... in these sentences the two strains were switched. It is the opposite of what is written.
Response 29: It has been corrected. Please see line 240-241, thank you.
Point 30: - line 260: please add “and Leuconostoc” after “Lactobacillus”
Response 30: It has been corrected. Please see line 254, thank you.
page 8
Point 31: - lines 269-271: “The Leuconostoc strain H52 supernatant-treated grapes significantly decreased in a downward trend with the control (P <0.05), however, the Lactobacillus strain F17 supernatant-treated grapes had the highest total phenols.” This sentence is too complicate and difficult to understand. Please reformulate.
Response 31: It has been corrected. Please see lines 262-265, thank you.
Point 32: - lines 278,285,290: please add “about” before CFU/ml values (i.e. “were about 2.14 CFU/ml”)
Response 32: It has been corrected. Please see lines 272, 279 and 284, thank you.
Point 33: - lines 280-281: please change “Lactobacillus strain F17” to “F17 and H52”
Response 33: It has been corrected. Please see line 274, thank you.
Point 34: - line282: please add a full stop after “20” and delete “and strain” before F17
Response 34: It has been changed. Please see line 276, thank you.
Point 35: - line 286: please add “and Leuconostoc strain H52” after “Lactobacillus strain F17”
Response 35: It has been corrected. Please see line 280, thank you.
Point 36: - lines 288-289: please change “a significant difference among all three groups during” to “a significant difference compared to the other two groups at”
Response 36: It has been corrected. Please see lines 282-283, thank you.
Point 37: - lines 290- 291: please delete “before the treatments were administered”
Response 37: It has been deleted. Please see lines 284, thank you.
Point 38: - line 291: please change “Lactobacillus” to “all”
Response 38: It has been corrected. Please see line 285, thank you.
Point 39: - line 292: please delete “throughout the study”
Response 39: It has been deleted. Please see line 285, thank you.
Point 40: - lines 295-296: please delete “the end of storage time, while the number of CB on the H52 supernatant-treated grapes were” and add “20 d: it was” after “2.51 log CFU/g at”
Response 40: It has been changed. Please see line 288, thank you.
Discussion
page 11
Point 41: -lines 364-368: complicated sentence. Please re-word. Preferably, split into two separate sentences.
Response 41: It has been corrected. Please see lines 356-359, thank you.
page12
Point 42: - lines 383-387: Actually, the pH also increased in control group. How do you explain it?
Response 42: It could be explained that organic acids, as the substrates for grape respiration, were gradually consumed with the extended storage time, so the pH of grapes also increased in the control group.
Point 43: - line 396: please change “inhibited the downward trend of” to “increased”
Response 43: It has been corrected. Please see line 388, thank you.
Point 44: - lines 401-404: complicated sentence. Please re-word into two separate sentences.
Response 44: It has been corrected. Please see lines 393-396, thank you.
Point 45: - lines 412-413: “The strain H52 supernatant-treatment significantly inhibited YAMs among all groups over the entire storage period and may be a result of production of an extracellular polysaccharide”. Actually, YAMs decreased only between 10 to 20 days...
Response 45: It has been corrected. Please see lines 404-405, thank you.
Point 46: - lines 426-428: please delete “The obtained result also was consistent with the strongest purchase intention assessment. Additionally, table grapes treated with the strain H52 supernatant received an evaluation for purchase intention as 'liked moderately'.”
Response 46: It has been deleted. Please see lines 418, thank you.
Conclusions
page13
Point 47: - line 456: please delete “of YAMs”. It's a repetition
Response 47: It has been deleted. Please see line 446, thank you.
Point 48: - lines 456-457: please delete “the supernatant of”
Response 48: We left this phrase in the conclusion section because it was important for the reader to get the correct understanding, thank you.
Reviewer 2 Report
In this manuscript, the authors revealed that treatment with culture supernatants from lactic acid bacteria significantly improved postharvest quality of table grapes. The authors analyzed many evaluation items related to the quality of grapes and this paper can be evaluated as a report of food preservation and quality control. However, this study lacks microbiological points of view. They only used supernatants of lactic acid bacteria and did not identified secondary metabolites. As to the microbiological analysis, they only determined CFU and did not analyzed community composition. I think that this paper is out of the journal's or section's scope and it will not be satisfied by the readers of Microorganisms. In my view, this manuscript is not suitable for publication in Microorganisms but it is ideally suited for publication in a journal related to food science or food preservation.
Author Response
In this manuscript, the authors revealed that treatment with culture supernatants from lactic acid bacteria significantly improved postharvest quality of table grapes. The authors analyzed many evaluation items related to the quality of grapes and this paper can be evaluated as a report of food preservation and quality control. However, this study lacks microbiological points of view. They only used supernatants of lactic acid bacteria and did not identified secondary metabolites. As to the microbiological analysis, they only determined CFU and did not analyzed community composition. I think that this paper is out of the journal's or section's scope and it will not be satisfied by the readers of Microorganisms. In my view, this manuscript is not suitable for publication in Microorganisms but it is ideally suited for publication in a journal related to food science or food preservation.
Response:
Dear Reviewer,
Thank you for your comments concerning our manuscript entitled “Secondary metabolites of Lactobacillus delbrueckii subsp. bulgaricus strain F17 as a biopreservative for improving postharvest quality of 'Red Globe' grapes”. (Manuscript ID: microorganisms-765723).
Lactobacillus delbrueckii subsp. bulgaricus strain F17 and Leuconostoc lactis H52, which were isolated from traditional fermented yak milk obtained in the Qinghai-Tibetan Plateau, are food-grade microorganisms. In addition, Lactobacillus and their metabolites are widely used to control foodborne pathogens and spoilage organisms in meat products to ensure food quality and safety. However, there has not been a report of an application of lactic acid bacterial products for the improvement of table grape quality during storage at 25 °C. Therefore, the focus of this study was on extending the shelf-life of fresh table grapes using a supernatant of Lactobacillus and Leuconostoc containing bacterial products as a potential biopreservative. Indeed, Lactobacillus delbrueckii subsp. bulgaricus strain F17 and Leuconostoc lactis H52 products have a great potential in the research and development of biological preservatives and in treatment to maintain postharvest quality and for controlling microbial safety of table grapes.
Thank you for your suggestion again,it is very important for us. Because of your suggestions, I have found areas of my current work that may be improved. In our future work we will be able to improve our research direction based on your suggestions!
Round 2
Reviewer 2 Report
The manuscript has been improved and can be acceptable.